# LOCALLY LINEAR UNSUPERVISED FEATURE SELECTION

## ABSTRACT

The paper, interested in unsupervised feature selection, aims to retain the features best accounting for the local patterns in the data. The proposed approach, called *Locally Linear Unsupervised Feature Selection* (LLUFS), relies on a dimensionality reduction method to characterize such patterns; each feature is thereafter assessed according to its compliance w.r.t. the local patterns, taking inspiration from *Locally Linear Embedding* (Roweis and Saul, 2000). The experimental validation of the approach on the scikit-feature benchmark suite demonstrates its effectiveness compared to the state of the art.

## 1 INTRODUCTION

Machine Learning faces statistical and computational challenges due to the increasing dimension of modern datasets. Dimensionality reduction aims at addressing such challenges through embedding the data in a lower dimensionality space, in an unsupervised (Karlen et al., 2008; Roweis & Saul, 2000; Tenenbaum et al., 2000; Wang et al., 2015) or supervised (Gaudel & Sebag, 2010; Fukumizu et al., 2004; Nazarpour & Adibi, 2015) way.

The requirement for *understandable* Machine Learning (Vellido et al., 2012; Doshi-Velez & Kim, 2017) however makes it desirable to achieve interpretable dimensionality reduction. In order to do so, the simplest way is to select a subset of the initial features, i.e. to achieve feature selection (FS), as opposed to generating compound new features from the initial ones, a.k.a. feature construction. For instance, determining the genes most important w.r.t. a given disease or the underlying generative model of the data can be viewed as the mother goal in bioinformatics (Guyon et al., 2002b; Libbrecht & Noble, 2015).

In the supervised ML setting, features are assessed and selected based on their relevance to the prediction goal (Guyon & Elisseeff, 2003; Sheikhpour et al., 2017; Chen et al., 2017). Unsupervised learning, aimed at making sense of the data, however constitutes a primary and most important task of ML, as emphasized by LeCun (2017), while supervised ML intervenes at a later stage of the data exploitation process.

Unsupervised FS approaches (He et al., 2005; Zhao & Liu, 2007; Cai et al., 2010; Li et al., 2012; Zhu et al., 2017) (more in section 2) essentially rely on the assumption that the data samples are structured in clusters, and use the cluster partition *in lieu* of labels, making it possible to fall down on supervised FS, and select the features most amenable to characterize and separate the clusters. A main limitation of this methodology is that clusters are bound to rely on some metric defined from the initial features (with the notable exception of Li et al. (2012)), although this metric can be arbitrarily corrupted based on irrelevant or random features. On the other hand, as far as one considers the unsupervised setting, a feature can hardly be considered irrelevant *per se*.

The main contribution of the paper is to address both limitations: the proposed approach, called *Locally Linear Unsupervised Feature Selection* (LLUFS) jointly determines patterns in the data, and features relevant to characterize these patterns. LLUFS is a 2-step process (Sec. 3): In a first step, a compressed representation of the data is built using Auto-Encoders (Vincent et al., 2008; Feng et al., 2014). In a second step, viewing the initial dataset as a high-dimensional embedding of the compressed dataset, each feature is scored according to its contribution to the reconstruction error of the embedding, taking inspiration from Locally Linear Embedding (Roweis & Saul, 2000; Saul & Roweis, 2003; Wang, 2012).

After describing the goals of experiments and the experimental setting used to validate the approach, extensively relying on the scikit-feature project (Li et al., 2017; SKf, 2018) (Sec. 4), the empirical validation is presented and discussed (Sec. 5), establishing the merits and discussing the weaknesses of the approach. The paper concludes with a discussion and some perspectives for further research.

## NOTATIONS

$X$ denotes the $m \times D$ data matrix. Row $X[i,:]$, also noted $x_i$ when no confusion is to fear, is the $i$-th sample ($x_i$ in $\mathbb{R}^D$). Column $X[:,j]$ is the $j$-th feature. For $j$ in $[[1, D]]$, $\mu_j$ and $\sigma_j$ respectively denote the mean and the standard deviation of the $j$-th feature on the dataset. $1$ denote the $m$-dimensional constant vector $[1, ..., 1]^t$.

Let $S$ denote an $m \times m$ similarity matrix ($S_{i,i} > 0$), with $\Delta$ the associated diagonal degree matrix ($\Delta_i = \sum_{i=1}^{m} S_{i,j}$). The un-normalized (resp. normalized) Laplacian associated to $S$ is defined as $L = \Delta - S$ (resp. $\widetilde{L} = \Delta^{\frac{-1}{2}} L \Delta^{\frac{-1}{2}}$). Two particular similarities will be considered in the following; the supervised similarity $SUP$, with $SUP_{i,j} = 0$ iff $x_i$ and $x_j$ do not belong to the same class, and $1/|C_i|$ if they both belong to class $C_i$ , and the unsupervised similarity $RBF$, with $RBF_{i,j} = exp\{-\frac{1}{\delta}\|x_i - x_j\|_2^2\}$, and $\delta$ a hyper-parameter of the method.

## 2 FORMAL BACKGROUND

Supervised FS aims to select a subset of features such that it maximizes the eventual classifier accuracy. Supervised FS algorithms divide into filters, wrappers and embedded methods. Filter methods (Yu & Liu, 2003; Senawi et al., 2017) operate at the data pre-processing stage, and are agnostic to the classifier algorithm. Wrappers methods (Kabir et al., 2010; Rodrigues et al., 2014) aim to determine the feature subset yielding a best accuracy when used within a specific classifier, through solving a black-box optimization problem. Embedded methods (Guyon et al., 2002a; Zhang et al., 2015) alternatively learn and use the learned hypothesis to prune/select the unpromising/promising features. Admittedly, wrapper and embedded approaches might produce a candidate feature set with moderate generality (being linked to a particular classifier) and moderate interpretability (with the retained features being good as a gang). Since this paper focuses on unsupervised and interpretable FS, only filter methods will thus be considered in the following.

An early supervised filter method based on the so-called Fisher score was introduced by Duda et al. (2000), independently ranking features according to their correlation with the labels.[1] A general limitation of such scores is that they achieve a myopic feature selection, with XOR problems − where all relevant features need to be taken into account simultaneously − as typical failure cases.

A prominent unsupervised filter approach is based on spectral clustering: data clusters are first built using some metric or similarity; thereafter supervised FS approaches are used with these clusters *in lieu* of classes (Von Luxburg, 2007; Binkiewicz et al., 2017). He et al. (2005) introduce the Laplacian score[1], where each feature score measures how well this feature accounts for the sample similarity. Interestingly, while the Fisher score is a particular case of Laplacian score using the $SUP$ similarity, the Laplacian score overcomes the myopic limitations of the Fisher score when using the $RBF$ similarity. The Laplacian score is also remotely related to the MaxVariance FS method (Kantardzic, 2003), selecting features with large variance for the sake of their higher representative power.

Also relying on spectral clustering is the SPEC approach (Zhao & Liu, 2007), proposing three scores respectively noted $\phi_1$, $\phi_2$ and $\phi_3$. SPEC relies on the core idea that relevant features be smooth w.r.t. the graph, i.e. slowly varying among samples close to each other. After the spectral clustering theory (Shi & Malik, 1997; Ng, 2001), considering eigenvectors $\xi_0, ..., \xi_{m-1}$ of the normalized Laplacian $\widetilde{L}$ (respectively associated with eigenvalues $\lambda_0 < \lambda_1 < ... < \lambda_{m-1}$), smooth features are aligned with the first eigenvectors, hence the score $\phi_1$:

$$\forall j \in [[1, D]], \phi_{1j} = \widehat{X[:,j]}^t \widetilde{L} \widehat{X[:,j]} \text{ where } \widehat{X[:,j]} = \Delta^{\frac{1}{2}} X[:,j]/\left\|\Delta^{\frac{1}{2}} X[:,j]\right\| \tag{1}$$

---

[1]For the sake of space limitations, formal definitions are reminded in Appendix 1.

Eigenvectors $\xi_0, ..., \xi_{m-1}$ of $\widetilde{L}$ define soft cluster indicators, and eigenvalues $\lambda_0 < \lambda_1 < ... < \lambda_{m-1}$ measure the separability of the clusters. The smaller $\phi_{1j}$, the more efficient the $j$-th feature is to separate the clusters.

As the first eigenvector $\xi_0 = \Delta^{\frac{1}{2}} 1$ does not carry any information, with $\lambda_0 = 0$, one might rather consider the projection of the feature vector $X[:,j]$ on the orthogonal space of $\xi_0$:

$$\forall j \in [[1,D]], \phi_{2j} = \frac{1}{1 - \langle \widehat{X[:,j]}, \xi_0 \rangle} \widehat{X[:,j]}^t \widetilde{L} \widehat{X[:,j]} \tag{2}$$

Finally, in the case where the target number of clusters $\kappa$ is known, only the top-$\kappa$ eigenvectors are considered, and score $\phi_3$ is defined as:

$$\forall j \in [[1,D]], \phi_{3j} = \sum_{k=1}^{\kappa-1} (2 - \lambda_k) \langle \widehat{X[:,j]}, \xi_k \rangle^2 \tag{3}$$

Features are ranked in ascending order for $\phi_1$ and $\phi_2$, and in descending order for $\phi_3$.

The above three scores measure the overall capacity of a feature to separate clusters, which might prove inefficient in multi-classes/multi clusters settings: a feature most efficient to separate a pair of clusters might have a mediocre general score. The Multi-Cluster Feature Selection (MCFS) (Cai et al., 2010) addresses this limitation by defining a score *per cluster*. Formally, the capacity of $X[:,j]$ to separate clusters is estimated through fitting the eigenvectors (reminding that $\xi_k$ is a soft indicator of the $k$-th cluster) up to a regularization term. Letting A[k,:] denote a vector in $\mathcal{R}^D$, with $\beta$ a regularization weight:

$$\forall k \in [[1,\kappa]], A[k,:] = \min_{B \in \mathcal{R}^D} \|\xi_k - XB\|_2^2 + \beta \|B\|_1 \tag{4}$$

The $L_1$ regularization term enforces the sparsity of $A[k,:]$, retaining only the features most relevant to this cluster. The overall MCFS score simply takes the maximum over all clusters of the absolute value of $A_{k,j}$:

$$\forall j \in [[1,D]], MCFS_j = \max_{k \in [1,\kappa]} |A_{k,j}| \tag{5}$$

A general limitation of the above scores is to rely on a similarity metric that can be arbitrarily corrupted by noisy features, potentially leading to irrelevant clusters and scores. This limitation is addressed by Li et al. (2012), introducing the Nonnegative Discriminative Feature Selection (NDFS) approach. NDFS jointly optimizes the $D \times D$ feature importance matrix $A$ together with a cluster indicator matrix $\xi$, with an $L_2$ regularization:

$$\xi^*, A^* = \arg\min_{\xi,A} Tr(\xi^t \widetilde{L} \xi) + \alpha(\|\xi - XA\|_F^2 + \beta \|A\|_1^2) \tag{6}$$

subject to $\xi$ orthogonal and semi-positive definite ($\xi^t \xi = I_D, \xi \geq 0$), with $\alpha, \beta$ regularization weights. Following Yu & Shi (2003), the first term is rewritten as :

$$Tr(\xi^t \widetilde{L} \xi) = \frac{1}{2} \sum_{i,j=1}^m S[i,j] \left\| \frac{\xi[i,:]}{\sqrt{\Delta_i}} - \frac{\xi[j,:]}{\sqrt{\Delta_j}} \right\|_2^2 \tag{7}$$

The minimization of Eq. (6) tends to enforce the intra-cluster similarity and the inter-cluster dissimilarity. The orthogonality and nonnegativity constraints on $\xi$ further enforce that each sample belong in exactly one cluster.

## 3 LOCALLY LINEAR UNSUPERVISED FEATURE SELECTION (LLUFS)

This section first presents LLE for the sake of self-containedness, before presenting and discussing LLUFS.

### 3.1 LOCALLY LINEAR EMBEDDING

LLUFS takes inspiration from the *Locally Linear Embedding* (LLE) defined by Roweis & Saul (2000); Saul & Roweis (2003). LLE relies on the so-called Johnson Lindenstrauss lemma (Larsen & Nelson, 2017):

**Johnson-Lindenstrauss Lemma** .

$\forall \epsilon \in ]0, 1[, \forall d > \frac{8ln(m)}{\epsilon^2}, \exists f : \mathbb{R}^D \to \mathbb{R}^d$ s.t. :

$\forall (i, j) \in [[1, m]]^2, (1 - \epsilon) \|x_i - x_j\|^2 < \|f(x_i) - f(x_j)\|^2 < (1 + \epsilon) \|x_i - x_j\|^2$ with $f$ being a linear mapping (composed only of translations, rotations and rescalings).

As this lemma guarantees the existence of a low-dimensional embedding approximately preserving the pairwise distances among the points, LLE (Roweis & Saul, 2000): i) defines the local structure of the $m$ data points $x_i \in \mathcal{R}^D$, through approximating each point as the barycenter of its $n_W$ nearest neighbors; ii) finds points $z_1, \ldots z_m$ in $\mathcal{R}^d$, with $d \ll D$, such that the $z_i$ satisfy the same local relationships as the $x_i$s. Formally, let $N(i)$ denote the set of indices of the $n_W$ nearest neighbors of $x_i$; weights $W_{i,j}$ such that they minimize the Euclidean distance

$$\|x_i - \sum_{j \in N(i)} W_{i,j} x_j\|$$

subject to $\sum_{j \in N(i)} W_{i,j} = 1$, $W_{i,j} \geq 0$ and $W_{i,j} = 0$ for $j \notin N(i)$. Note that $W$ is invariant under rotation, translation or homothety on the dataset $X$: it captures the local structure of the $x_i$s. The LLE dimensionality reduction thus proceeds by finding another set of points $z_i$s in $\mathcal{R}^d$, such that they satisfy the local relationships expressed by $W$:

$$Z^* = \underset{Z \in \mathbb{R}^{m \times d}}{\arg \min} \|Z - WZ\|_F^2 \tag{8}$$

## 3.2 Overview of LLuFS

While LLE is performed as a dimensionality reduction technique, its principle is general: after the local structure of the data has been captured through matrix $W$, this matrix can be used to transport the data from any source to target representation.

As our goal is to achieve feature selection, we implicitly assume that the $X$ data live in a low-dimension space. Accordingly, the proposed LLuFS approach proceeds by: i) finding a low-dimension representation of the data in $\mathcal{R}^d$; ii) characterizing the matrix $W$ capturing the data structure *in this low-dimension representation*; iii) using $W$ to assess the initial features, as detailed below.

### 3.2.1 Dimensionality reduction of $X$

This step can be achieved using linear or non-linear approaches, ranging from PCA (Wold et al., 1987) and SVD (Deerwester et al., 1990) to Isomap (Tenenbaum et al., 2000) or t-SNE (Maaten & Hinton, 2008). For the sake of generality and robustness, LLuFS uses the non-linear Stacked Denoising AutoEncoder neural networks (SDAE) (Vincent et al., 2010; Feng et al., 2014), meant to achieve a non-linear compression robust w.r.t. input noise.

Let $Z$ denote the $(m, d)$ data obtained from $X$ through this dimensionality reduction. For each $i = [[1, m]]$, let $N(i)$ be the set of $n_W$ nearest neighbors of $z_i$, with $W$ the $(m, m)$ matrix minimizing $\|Z - WZ\|$ under the positivity and sum-to-1 constraints defined in Sec. 3.1.

### 3.2.2 Distorsion score

Let us consider the $Z$ as the "true" data, with the $X$ as an inflated and corrupted image of the $Z$. The overall loss of information from $Z$ to $X$ is measured as $\|X - WX\|_F^2$. Most interestingly, this overall loss of information can be decomposed with respect to examples, with:

$$Err(x_i) = \left( x_i - \sum_{j \in N(i)} W_{i,j} x_j \right)^2$$

and with respect to the initial features, with:

$$\text{Distorsion}(X[:, j]) = \sum_{i=1}^{m} \left( X[i, j] - \sum_{k \in N(i)} W_{i,k} X[k, j] \right)^2$$

The distorsion associated to the $j$-th feature is thus interpreted in terms of how much this feature is corrupted with respect to the "true" local structure of the data, defined from the $Z$s. The features with lowest distorsion thus are deemed the most representative of the data. Note that, although the distorsion score is defined for each initial feature, it might implicitly take into account the global structure of the data, captured by the $W$.

### 3.3 DISCUSSION

One weakness of the method is that the distorsion scores depend on the latent representation produced by the auto-encoder, which might be biased due to the redundancy of the initial features; typically, duplicating an initial feature will entail that the latent representation is more able to express this feature, mechanically reducing its distorsion score. For this reason, a preliminary step is to detect and reduce the redundancy of the initial features.

In order to do so, LLUFS i) normalizes the initial features (with zero mean and unit variance); ii) uses Agglomerative Hierarchical feature clustering (Krier et al., 2007; VanDijck & VanHulle, 2006), using a high number of clusters $n_c$ ($n_c = \frac{3}{4}D$ in the experiments); iii) selects one feature per cluster (the nearest one to the cluster mean); iv) apply the auto-encoder on the pruned data.

Further work is concerned with taking into account the feature redundancy within the AE loss.

A second limitation is due to the sensitivity of the distorsion score to the feature distribution. Typically, while a constant feature carries no information, its distorsion is null. Likewise, the distorsion of discrete features depends on their being balanced. In order to alleviate this issue, the reliability of the distorsion associated to each feature is measured through an empirical $p$-value (Stoppiglia et al., 2003). Given a $p$-value threshold $\tau$, $\lfloor 1/\tau \rfloor$ copies of each feature are generated and independently shuffled. The feature distorsion is deemed relevant iff it is lower than the distorsion of all shuffled copies.

---

**Algorithm 1:** Locally Linear Unsupervised Feature Selection (LLUFS)

---

1 LLUFS (empirical p-value threshold $\tau$, number of clusters $n_c$, embedding dimension $d$, number of neighbors $n_W$);
**Input** : $X, n_c, \tau, d, n_W$
2 Normalize features to zero mean and unit variance.
3 Perform Agglomerative Hierarchical Feature Clustering, producing $X_{filtered} \in \mathbb{R}^{m \times n_c}$.
4 Train a SDAE on $X_{filtered}$ to produce compressed representation $Z$.
5 Solve $W = \arg\min \|Z - WZ\|_F^2$ subject to the positivity and sum-to-1 constraints.
6 **for** $j \in [[1, n_c]]$ **do**
7 $\quad$ Compute $Distorsion(X_{filtered}[:, j]) = \sum\limits_{i=1}^{m} |X_{filtered}[i, j] - \sum\limits_{l=1}^{m} W[i, l] X_{filtered}[l, j]|$
8 **end**
9 Initialize set of candidates $Cand$ to all features, ranked in ascending order w.r.t. distorsion. Initialize selection subset $Sel = \emptyset$.
10 **while** $|Cand| > 0$ **do**
11 $\quad$ Create $\lfloor \frac{1}{\tau} \rfloor$ random permutations $\sigma_0, ..., \sigma_{\lfloor \frac{1}{\tau} \rfloor}$ of the first feature $f_0$ in $Cand$.
12 $\quad$ **for** $k \in [[1, \lfloor \frac{1}{\tau} \rfloor]]$ **do**
13 $\quad\quad$ Compute $Distorsion(\sigma_k) = \sum\limits_{i=1}^{m} |f_0[\sigma_k(i)] - \sum\limits_{l=1}^{m} W[i, l] f_0[\sigma_k(l)]|$
14 $\quad$ **end**
15 $\quad$ **if** $Distorsion(f_0) < \min_{k \in [[1, \lfloor \frac{1}{\tau} \rfloor]]} Distorsion(\sigma_k)$ **then**
16 $\quad\quad$ $Sel \leftarrow Sel \cup f_0$.
17 $\quad$ **end**
18 $\quad$ $Cand \leftarrow Cand - f_0$.
19 **end**
20 Return $Sel$

---

## 4 EXPERIMENTAL SETTING

### 4.1 GOALS OF EXPERIMENTS

The main goal of the experimental validation is to assess LLUFS compared to state of the art un-supervised feature selection approaches. The performance assessment commonly falls back on the supervised setting, where the indicator is the predictive accuracy of a classifier trained from the selected features, where the number of selected features ranges from 1 to $D$ (Chen et al., 2017).[2] For the sake of clarity and to sidestep issues related to classifier hyper-parameter tuning, the classifier considered in the following is the 1-nearest neighbor classifier. Secondly (Q2), the respective impacts of both LLUFS ingredients, the feature clustering pre-processing, and the proper FS mechanism, are assessed. Thirdly (Q3), experiments will investigate the robustness of the proposed approach specifically w.r.t. XOR concepts (Sec. 2).

### 4.2 BASELINES AND BENCHMARKS

The experimental setting extensively relies on the scikit-feature project (Li et al., 2017; SKf, 2018), defining a *de facto* standard for FS approaches through algorithm implementations and datasets (Chen et al., 2017; Zadeh et al., 2017). Five baseline algorithms are considered: Laplacian Score (LAP) (He et al., 2005), Spectral Feature Selection (SPEC, considering the $\phi_1$ score, Sec. 2) (Zhao & Liu, 2007), Multi-Cluster Feature Selection (MCFS) (Cai et al., 2010), and Non-Negative Discriminative Feature Selection (NDFS) (Li et al., 2012); the fifth and last baseline (RANDOM), aimed to assess the impact of the only feature pre-processing in LLUFS (Q2), is defined by uniformly selecting the features after the feature clustering process (Alg. 1). 7 benchmark datasets from SKf (2018) are considered (Appendix 2): 6 datasets in the domains of image and bioinformatics, and the Madelon XOR problem (Guyon & Elisseeff, 2003) to empirically investigate (Q3). The number of features range from 500 to 10,000; the number of classes range from 2 to 11, and the number of examples is less than 200 (except for Madelon, with 2,000 examples). As discussed in Sec. 3.3, only datasets with continuous features are considered. All features are normalized (zero mean, unit variance).

### 4.3 HYPER-PARAMETERS

LLUFS involves three hyper-parameters: the number $n_c$ of clusters used in the feature-preprocessing; the methodology used to build the latent representation; and the number of neighbors $n_W$ considered in Sec. 3 to build the $W$ matrix.
$n_c$ is set to $\frac{3}{4}D$ for all datasets. The methodology used to build the latent representation is a 5-layer stacked denoising auto-encoder (Vincent et al., 2010) with architecture $D - \frac{D}{2} - \frac{D}{4} - \frac{D}{8} - \frac{D}{4} - \frac{D}{2} - D$ with $tanh$ activation function, trained to minimize the MSE loss for $10^2$ epochs with a $10^{-3}$ learning rate. The denoising process uniformly selects 20% of the features and sets them to 0 for each example. $n_W$ is set to 6 for all datasets, considering the small number of samples.

## 5 EMPIRICAL VALIDATION

This section reports and discusses the comparative performance of all unsupervised FS methods, in view of the experiment goals.

---

[2]Another option is to cluster the samples w.r.t. the selected features, and consider the normalized mutual information between class labels and cluster labels as a performance indicator. Preliminary experiments however show that this indicator is hardly sensitive to the considered FS method.

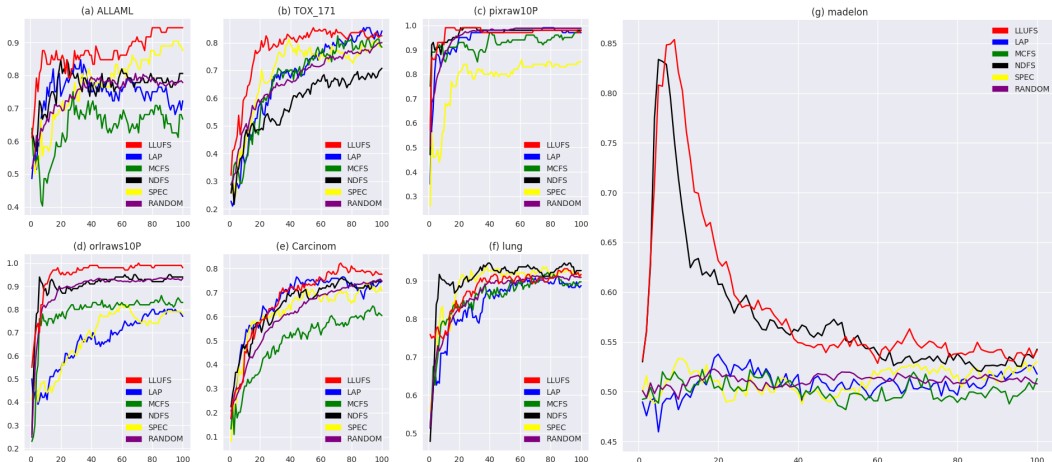

Figure 1: Assessment of unsupervised FS algorithms LLUFS, LAP, SPEC, MCFS, NDFS and RAN-
DOM: predictive accuracy of 1-nn classifier *vs* number $d$ of selected features, on datasets MADELON,
ALLAML, CARCINOM, LUNG, ORLRAWS10P, PIXRAW10P, TOX171
.

## 5.1 COMPARISON ON BIOLOGICAL AND IMAGE DATASETS

Fig. 1 displays the performance curve (1-nearest neighbor accuracy) *vs* the number $d$ of selected
features.[3] On datasets ALLAML and TOX171 (Fig.1 (a) and (b)), LLUFS dominates all other meth-
ods over the whole learning curve. On ALLAML, both LAP and NDFS do much better than MCFS,
suggesting that the feature clusters are not much relevant to the classification task. SPEC shows
a robust performance after sufficiently many features have been selected ($d > 50$). On TOX171,
both LAP and MCFS do much better than NDFS, suggesting that the feature set presents a complex
cluster structure (captured by NDFS) misleading to the classification task. Likewise, SPEC yields
good results after the beginning of the curve ($d > 20$). In both cases, RANDOM is significantly
outperformed, suggesting that quite a few feature clusters are irrelevant to the prediction task.

On datasets PIXRAW10P and ORLRAWS10P (Fig.1 (c) and (d)), LLUFS is dominated by NDFS at
the beginning of the curve ($d < 5$ for PIXRAW10P and $d < 10$ for ORLRAWS10P); it thereafter
dominates all other algorithms on ORLRAWS10P (resp. dominates the others then reaches the same
nearly maximal performance as all other algorithms on PIXRAW10P). The relative comparative
weakness of LLUFS at the very beginning of the curve is interpreted as LLUFS capturing patterns
related to subsets of features (as the latent representation is bound to globally account for the initial
features). The importance of a feature standalone thus can hardly be accounted for, contrasting with
NDFS. On both datasets, RANDOM yields a decent performance (ranking 2nd or 3rd, especially for
high values of $d$), which suggests that a main issue with those image datasets is the redundancy of
the features.

On CARCINOM (Fig.1 (e)), LLUFS is dominated by all algorithms but MCFS at the beginning of
the curve ($d < 20$); it then catches up and dominates the other algorithms for $d > 60$. The best
algorithm on this dataset is LAP, suggesting that the cluster structure defined from all features is
relevant to the prediction task, which might explain why LLUFS and MCFS, more sensitive to local
patterns, are outperformed.

On LUNG (Fig.1 (f)), LLUFS is consistently dominated by NDFS and SPEC over the whole learning
curve and performs similarly as RANDOM, suggesting that the compressed representation learned
by the neural network does not accurately represent data structure.

---

[3]The variance of the predictive accuracy over 25 independent runs of the RANDOM baseline is circa 0.02
for $d < 5$, $5 * 10^{-3}$ for $d = 20$); the confidence bars are omitted in the figures for the sake of readability.

## 5.2 The XOR problem

As said, the Madelon problem (Appendix 2) is chosen to investigate the comparative performances of unsupervised FS methods in the case where, by construction, FS based on independent feature scores are bound to fail. Two clusters of methods are clearly seen on this problem (Fig. 1 (g)). Most methods fail to do better than random selection, due to the low signal to noise ratio (96% of the features being pure noise), adversely affecting spectral clustering methods and clusters based on Laplacian eigenvectors. NDFS does much better, as it simultaneously learns cluster indicators and feature relevance. LLUFS does even better as the latent representation tends to highlight the data patterns, and the distorsion score measures whether a feature is relevant to these patterns. The drop of performance of NDFS and LLUFS, after the number of selected features goes beyond the number of relevant features (20), is blamed on the addition of noise features perturbing the metric and misleading the 1-nn classifier.

## 6 Conclusion and future work

A novel approach to unsupervised feature selection has been proposed in this paper, with a proof of concept of its empirical merits. The core idea is to find an "oracle" representation of the data, and to consider the actual data as an inflated and corrupted image of the oracle data. The quality of each feature is thereafter assessed depending on how it contributes to the loss of information between the "oracle" and the actual data.

A first perspective for further research, taking inspiration from NDFS, is to allow a feature to be partially relevant, e.g. through considering the quantiles of its distorsion. A second perspective is to integrate the feature redundancy in the auto-encoder loss, to decrease the bias in favor of redundant features. The approach will also be extended to supervised feature selection.

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

## APPENDIX 1

### FISHER SCORE

Denoting $c$ the number of classes, $n_i$ the number of samples in the $i$-th class, $\mu_{i_j}$ and $\sigma_{i_j}$ respectively the mean and the standard deviation of the $j$-th feature on the $i$-th class, the Fisher score is defined as:

$$\forall 1 \leq j \leq D, F_j = \frac{\sum\limits_{i=1}^{c} n_i(\mu_{i_j} - \mu_j)^2}{\sum\limits_{i=1}^{c} n_i\sigma_{i_j}^2} \tag{9}$$

### LAPLACIAN SCORE

With same notations,

$$\forall 1 \leq j \leq D, L_j = \frac{1}{\sigma_j} \sum\limits_{i,k=1}^{m} (X_{i,j} - X_{k,j})S_{i,k} \tag{10}$$

with $\sigma_j$ the standard deviation of the $j$-th feature and $S_{i,k}$ the similarity of the $i$-th and $k$-th examples.

Note that $L_j$ can be rewritten using the Laplacian matrix:

$$L_j = \frac{\widetilde{X[:,j]}^t L \widetilde{X[:,j]}}{\widetilde{X[:,j]}^t \Delta \widetilde{X[:,j]}}$$

with $\widetilde{X[:,j]} = X[:,j] - \frac{X[:,j]^t\Delta 1}{1^t\Delta 1}1$.

Furthermore, the Fisher score is a particular case of the Laplacian score:

$$L_j = \frac{1}{1 + F_j}$$

for similarity $S$ set to the SUP similarity.

## APPENDIX 2

### SUMMARY OF DATASETS CONSIDERED

| Name | # Instances | # Features | # Classes | Data type |
|------|-------------|------------|-----------|-----------|
| Madelon | 2000 | 500 | 2 | Artificial XOR |
| orlraws10P | 100 | 10304 | 10 | Face Image |
| pixraw10P | 100 | 10000 | 10 | Face Image |
| ALLAML | 72 | 7129 | 2 | Biological |
| Carcinom | 174 | 9182 | 11 | Biological |
| lung | 203 | 3312 | 5 | Biological |
| TOX_171 | 171 | 5748 | 4 | Biological |

Madelon, an artificial XOR-like dataset, was created for the Feature Selection Challenge of NIPS2003 (Guyon, 2003). It involves 5 relevant features, the values of which are combined along two XOR concepts to define the positive and negative classes. Specifically, each class includes $2^5$ Gaussian clusters, placed on the vertices of a hypercube. The relevant 5 features are duplicated, combined and perturbed to obtain 15 "distractor" features. 480 standard normal noise features are then added. Most classifiers (SVMs with linear, polynomial or Gaussian kernels; NNs with up to 10 layers) using all features yields the same $50\%$ accuracy as random prediction, indicating that efficient FS is compulsory.

## APPENDIX 3

### SUMMARY OF EXPERIMENTAL RESULTS

Table 1: Summary of predictive accuracy for each dataset and FS algorithm along the learning curve, resp. for 2/5/10/20/50/100 features selected

|           | LLUFS                          | LAP                            | SPEC                           |
|-----------|--------------------------------|--------------------------------|--------------------------------|
| ALLAML    | 0.65/0.75/0.86/0.85/0.85/0.94  | 0.54/0.60/0.70/0.74/0.75/0.72  | 0.61/0.56/0.58/0.69/0.76/0.87  |
| Carcinom  | 0.25/0.30/0.45/0.57/0.74/0.78  | 0.22/0.31/0.50/0.58/0.75/0.75  | 0.15/0.28/0.46/0.58/0.71/0.71  |
| Lung      | 0.75/0.77/0.75/0.82/0.91/0.92  | 0.56/0.63/0.71/0.79/0.88/0.89  | 0.60/0.69/0.78/0.90/0.92/0.91  |
| Madelon   | 0.56/0.81/0.82/0.63/0.54/0.54  | 0.48/0.46/0.48/0.54/0.51/0.52  | 0.52/0.49/0.53/0.51/0.49/0.53  |
| orlraws10P| 0.62/0.74/0.91/0.95/0.98/0.98  | 0.41/0.39/0.44/0.54/0.66/0.77  | 0.45/0.40/0.45/0.53/0.77/0.78  |
| pixraw10P | 0.87/0.88/0.92/0.98/0.97/0.98  | 0.58/0.88/0.87/0.95/0.98/0.97  | 0.56/0.47/0.57/0.81/0.80/0.85  |
| TOX171    | 0.41/0.46/0.60/0.77/0.82/0.82  | 0.21/0.29/0.36/0.58/0.71/0.84  | 0.26/0.35/0.42/0.65/0.77/0.78  |
|           | MCFS                           | NDFS                           | RANDOM                         |
| ALLAML    | 0.55/0.55/0.49/0.58/0.64/0.67  | 0.61/0.64/0.68/0.85/0.78/0.81  | 0.54/0.60/0.65/0.70/0.78/0.78  |
| Carcinom  | 0.22/0.18/0.31/0.37/0.53/0.60  | 0.28/0.37/0.47/0.56/0.71/0.75  | 0.23/0.36/0.36/0.48/0.65/0.75  |
| Lung      | 0.58/0.67/0.79/0.83/0.87/0.90  | 0.56/0.84/0.90/0.89/0.94/0.93  | 0.58/0.70/0.75/0.82/0.89/0.91  |
| Madelon   | 0.49/0.49/0.51/0.51/0.49/0.51  | 0.56/0.83/0.72/0.61/0.57/0.54  | 0.49/0.51/0.49/0.52/0.52/0.51  |
| orlraws10P| 0.25/0.56/0.75/0.77/0.81/0.83  | 0.50/0.79/0.89/0.88/0.93/0.94  | 0.41/0.68/0.81/0.89/0.92/0.93  |
| pixraw10P | 0.89/0.91/0.86/0.93/0.93/0.98  | 0.92/0.90/0.93/0.97/0.98/0.98  | 0.68/0.77/0.88/0.96/0.98/0.99  |
| TOX171    | 0.30/0.35/0.36/0.53/0.74/0.78  | 0.30/0.33/0.46/0.49/0.62/0.71  | 0.30/0.38/0.49/0.58/0.70/0.80  |

