# OpenReview forum: "Locally Linear Unsupervised Feature Selection"
_ICLR.cc/2019/Conference_

### Official Review · AnonReviewer1 · 2018-10-31
**The paper lacks a solid motivation**

**Rating:** 3
**Confidence:** 5

**Review:**

Summary: The paper proposes the LLUFS method for feature selection. The idea is to first apply a dimensionality reduction method on the input data X to find a low-dimensional representation Z. Next, each point in Z is represented by a linear combination of its nearest neighbors by finding a matrix W which minimizes || Z  - WZ||. Finally, these weights are used to asses the distortion of every feature in X by considering the reconstruction loss in the original space.

Comments: There are multiple shortcomings in the motivation of the approach. First, the result of the dimensionality reduction drastically depend on the method used. It is well known that every DR method focuses on preserving certain properties of the data. For instance, PCA preserves the global structure while t-SNE works locally, maximizing the recall [1]. The choice of the DR method should justify the underlying assumption of the approach. I expect that the results of the experiments to change drastically by changing the DR method.

Second, the LLE method is based on the assumption that if the high-dimensional data is locally linear, it can be projected on a low-dimensional embedding which is also locally linear. Transitioning from a locally linear high-dimensional data to a lower dimension makes sense because there exists higher degree of freedom in the higher dimension. However, making this assumption in the opposite direction is not very intuitive. Why would the features that do not conform to the local linearity of the low-dimensional structure (which itself is obtained via a non-linear mapping) are insignificant?

Finally, there are no theoretical guarantees on the performance of the method. Is there any guarantee that, e.g. given one noisy feature in high dimension, the method will find that feature, etc.?

Minor: what is the complexity of the method compared to the competing methods? What is the runtime? Is this a practical approach on large datasets?

Overall, I do not agree with the assumptions of the paper nor convinced with the experimental study. Therefore, I vote for reject.

[1] Venna et al. "Information retrieval perspective to nonlinear dimensionality reduction for data visualization." Journal of Machine Learning Research 11, no. Feb (2010): 451-490.

---

> ### Author Response · Authors · 2018-11-19
> **Response to Reviewer1**
>
> Thank you for your review.
>
>
> Q1: "First, the result of the dimensionality reduction drastically depend on the method used.
> It is well known that every DR method focuses on preserving certain properties of the data.
> For instance, PCA preserves the global structure while t-SNE works locally, maximizing the recall [1].
> The choice of the DR method should justify the underlying assumption of the approach.
> I expect that the results of the experiments to change drastically by changing the DR method."
>
> A1: You are right, the result depends on the DR method. However:
> i) a linear DR does not work (as shown on the toy XOR example)
> ii) non-linear DR methods (Isomap, t-SNE, MDS, LLE) rely on the local Euclidean distance in the original space, that might be arbitrarily corrupted by random features.
> iii) The non-linear DR method in LLUFS (denoising auto-encoder with D-D/2-D/4-D/8-D/4-D/2-D neural architecture) yields stable results (e.g. 98% same features are in the top 100 selected features for all datasets w.r.t. different initialisations).
>
> Q2: "The LLE method is based on the assumption that if the high-dimensional data is locally linear,
> it can be projected on a low-dimensional embedding which is also locally linear.
> Transitioning from a locally linear high-dimensional data to a lower dimension makes sense because there exists higher degree of freedom
> in the higher dimension. However, making this assumption in the opposite direction is not very intuitive.
> Why would the features that do not conform to the local linearity of the low-dimensional structure (which itself is obtained via a non-linear mapping) are insignificant?"
>
> A2: The approach assumes that
> * X (the initial data in D dimensions) can be mapped onto Z (latent space in D/8 dimensions) with no or little loss of information;
> * From Z, the idea is to find among X_sub (all datasets defined from X by selecting a subset of features) the best mapping in the LLE sense. From Z (dimension D/8) to X_sub, the decrease in dimensionality is still high (the evaluation considers the selected top-100 features, with 100 << D/8 except for Madelon).
>
>
> Q3: "Finally, there are no theoretical guarantees on the performance of the method. Is there any guarantee that, e.g. given one noisy feature in high dimension, the method will find that feature, etc.?"
>
> A3: You are right, there is no theoretical guarantees for LLUFS. To our best knowledge, the evaluation of all unsupervised FS methods (including ours) is based on a supervised setting.
>
> Q4 : "Minor: what is the complexity of the method compared to the competing methods? What is the runtime? Is this a practical approach on large datasets?"
>
> A4 : Theoretically: Besides the complexity of learning the Auto-Encoder, the time complexity of the prior agglomerative hierarchical clustering is O(D**2) with D the number of features (up to logarithmic terms). This complexity motivates the extension proposed in the paper, to use the feature correlation within the Auto-Encoder loss to deal with redundancy (section 3.3).
> The time complexity of the nearest neighbor search is O(D/8 n**2) with D/8 the dimension of Z and n the number of points.
> The time complexity of computing the W matrix is  O(D/8 n k**3) with k the number of neighbors set to 6 in all problems.
> Empirically, LLUFS is slower than LAP, SPEC, MCFS and faster than NDFS. On dataset lung (203 points, 3312 features), the respective runtimes (on a single 2.67 Ghz CPU core) are :
> * 0.2 seconds for LAP.
> * 1.6 seconds for SPEC.
> * 24.5 seconds for MCFS.
> * 114.4 seconds for LLUFS (*)
> * 131.0 seconds for NDFS.
> (*) 24.4 seconds for the agglomerative clustering; 77.7 seconds for training the AutoEncoder; 12.3 seconds for the distorsion step.
>
> On dataset pixraw10P (100 points, 10 000 features) :
> *0.3 seconds for LAP.
> *1.8 seconds for SPËC.
> *258 seconds for MCFS.
> *930 for LLUFS (*)
> *1646 seconds for NDFS.
> (*) 614.6 seconds for the agglomerative clustering + 300 seconds for training the AutoEncoder + 15.9 seconds for the distorsion step.

---

### Official Review · AnonReviewer2 · 2018-11-01
**Locally Linear Unsupervised Feature Selection**

**Rating:** 6
**Confidence:** 2

**Review:**

In this paper, the authors presented Locally Linear Unsupervised Feature Selection (LLUFS), where a dimensionality reduction is first performed to extract data patterns, which are used to evaluate compliance of features to the patterns, applying the idea of Locally Linear Embedding.

1. This work basically assumes that the dataset is (well) clustered. This might be true for most real world dataset, but I believe the degree of clustered-ness may vary by dataset. It will be nice to discuss effect of this. For example, if most data points are concentrated on a particular area not being well clustered, how much this approach get affected? If possible, it will be great to formulate it mathematically, but qualitative discussion is still useful.

2. For the dimension reduction, the authors used autoencoder neural network only. What about other techniques like PCA or SVD? Theoretical and experimental comparison should be interesting and useful.

3. This paper is well-written, clearly explaining the idea mathematically. It is also good to mention limitation and future direction of this work. It is also good to cover a corner case (XOR problem) in details.

4. Minor comments:
 - Bold face is recommended for vectors and matrices. For instance, 1 = [1, 1, ..., 1]^T, where we usually denote the left-hand 1 in bold-face.
 - It seems x_j is missing in Johnson-Lindenstrauss Lemma formula. As it is, \sum_j W_{i,j} is subject to be 1, so the formula does not make sense.

---

> ### Author Response · Authors · 2018-11-19
> **Response to Reviewer2**
>
> Thank you for your review.
>
> Q1 : "This work basically assumes that the dataset is (well) clustered. This might be true for most real world datasets,
> but I believe the degree of clustered-ness may vary by dataset. It will be nice to discuss effect of this.
> For example, if most data points are concentrated on a particular area not being well clustered,
> how much this approach get affected? If possible, it will be great to formulate it mathematically, but qualitative discussion is still useful."
>
> A1 : Given the absence of label information, unsupervised FS algorithms rely on the assumption that there is some sort of "intrisic structure" to the data.
> Unsupervised approaches [1,2,3,4,5] assume that there are some clusters, which can be well-separated by an appropriate feature subset.
> As these clusters are defined in the initial feature space, they depend on the Euclidean distance which is arbitrarily corrupted from irrelevant features
> (except for [5] , which iteratively learns a new distance during selection).
>
> LLUFS proposes another strategy:
> * An auto-encoder achieves the non-linear dimensionality reduction and constructs features, defining a compressed version Z of the initial data X;
> * We now search for the subset of initial features, defining X_sub such that, if we applied LLE dimensionality reduction on Z, X_sub would be a perfect candidate (in the sense of preserving the local structure defined from W, with Z ~= WZ).
> * The gain is that the combinatorial optimization problem of finding the best subset of features of size d can be solved in a straightforward way as the score of each feature is its distorsion: ranking the initial features by increasing distorsion, the optimal set of features is the top d features.
>
>
> Q2 : "For the dimension reduction, the authors used autoencoder neural network only. What about other techniques like PCA or SVD?"
>
> A2 : Non separable clusters (the XOR problem) cannot be captured from a linear dimensionality reduction (PCA, SVD) method.
> It is true that we could have used other non-linear dimensionality reduction methods (Isomap or MDS) to define a latent representation, instead of Auto-Encoder. However, Isomap and MDS depend on the Euclidean distance in the initial feature space, thus with same weakness as said in A1.
>
>
> "It seems x_j is missing in Johnson-Lindenstrauss Lemma formula."
>
> You are right. Thank you. We fixed the typo.
>
> [1] Cai et al. (2010) "Unsupervised Feature Selection for Multi-Cluster Data",
> [2] Li et al. (2012) "Unsupervised feature selection using non-negative spectral analysis"
> [3] Li et al. (2014) "Clustering-guided sparse structural learning for unsupervised feature selection",
> [4] Shi et al. (2014) "Robust spectral learning for unsupervised feature selection"
> [5] Nie et al. (2016) "Unsupervised Feature Selection with Structured Graph Optimization"

---

### Official Review · AnonReviewer3 · 2018-11-07
**Locally Linear Unsupervised Feature Selection**

**Rating:** 4
**Confidence:** 5

**Review:**

This paper focuses on the problem of unsupervised feature selection, and proposes a method by exploring the locally linear embedding. Experiments are conducted to show the performance of the proposed locally linear unsupervised feature selection method. There are some concerns to be addressed.

First, the novelty and motivation of this paper is not clear. This paper seems to directly use one existing dimensionality reduction method, i.e., LLE, to explore the local structure of data. Why uses LLE rather than other methods such as LE? What are the advantages?

Second, in Section 3.3, authors state that the method might be biased due to the redundancy of the initial features. To my knowledge, there are some unsupervised feature selection to explore the redundancy of the initial features, such as the extended work of f Li et al. (2012) "Unsupervised Feature Selection via Nonnegative Spectral Analysis and Redundancy Control".

Third, how about the computational complexity of the proposed method? It is better to analyze it theoretically and empirically.

Finally, the equation above Eq. 8 may be wrong.

---

> ### Author Response · Authors · 2018-11-19
> **Response to Reviewer3**
>
> Thank you for your review and for the reference.
>
> Q1: "This paper seems to directly use one existing dimensionality reduction method, i.e., LLE, to explore the local structure of data"
>
> A1: Actually, LLUFS uses two dimensionality reduction approaches in complementary ways, along a 2-step process:
> * An auto-encoder achieves the non-linear dimensionality reduction and constructs features, defining a compressed version Z of the initial data X;
> * We now search for the subset of initial features, defining X_sub such that, if we applied LLE dimensionality reduction on Z, X_sub would be a perfect candidate (in the sense of preserving the local structure defined from W, with Z ~= WZ).
> * The gain is that the combinatorial optimization problem of finding the best subset of features of size d can be solved in a straightforward way as the score of each feature is its distorsion: ranking the initial features by increasing distorsion, the optimal set of features is the top d features.
>
>
> Q2:  "Why uses LLE rather than other methods such as LE? What are the advantages?"
>
> A2: Linear embedding can hardly be used in the first step if we want to capture non-separable patterns (e.g. XOR) in the initial representation.
> As for the second step, prior work such as [1,2,3] indicate that in order to be efficient, feature scoring must reflect data structure on a local scale.
> This observation motivates using the proposed distorsion score over global-scale methods such as PCA.
>
> Q3: "Authors state that the method might be biased due to the redundancy of the initial features.
> To my knowledge, there are some unsupervised feature selection to explore the redundancy of the initial features, such as the extended work of Li et al. (2012) "Unsupervised Feature Selection via Nonnegative Spectral Analysis and Redundancy Control"."
>
> A3: The authors of (Li et al., 2015) improve on NDFS [4] through an additional term on the feature importance matrix, penalizing the selection of correlated features.
> At the moment, feature redundancy is taken into account in LLUFS prior to launching the Auto-Encoder: using the feature correlation within the Auto-Encoder loss is a perspective for further work (section 3.3).
>
>
> Q4: "How about the computational complexity of the proposed method?"
>
> A4: Theoretically: Besides the complexity of learning the Auto-Encoder, the time complexity of the prior agglomerative hierarchical clustering is O(D**2) with D the number of features (up to logarithmic terms). This complexity motivates the proposed extension (Q3).
> The time complexity of the nearest neighbor search is O(D/8 n**2) with D/8 the dimension of Z and n the number of points.
> The time complexity of computing the W matrix is  O(D/8 n k**3) with k the number of neighbors set to 6 in all problems.
> Empirically, LLUFS is slower than LAP, SPEC, MCFS and faster than NDFS. On dataset lung (203 points, 3312 features), the respective runtimes (on a single 2.67 Ghz CPU core) are :
> * 0.2 seconds for LAP.
> * 1.6 seconds for SPEC.
> * 24.5 seconds for MCFS.
> * 114.4 seconds for LLUFS (*)
> * 131.0 seconds for NDFS.
>
> (*) 24.4 seconds for the agglomerative clustering; 77.7 seconds for training the AutoEncoder; 12.3 seconds for the distorsion step.
>
> On dataset pixraw10P (100 points, 10 000 features) :
> *0.3 seconds for LAP.
> *1.8 seconds for SPËC.
> *258 seconds for MCFS.
> *930 for LLUFS (*)
> *1646 seconds for NDFS.
> (*) 614.6 seconds for the agglomerative clustering + 300 seconds for training the AutoEncoder + 15.9 seconds for the distorsion step.
>
> Q5 "Finally, the equation above Eq. 8 may be wrong."
> You are right. Thank you. We fixed the typo.
>
> [1] Cai et al. (2010) "Unsupervised Feature Selection for Multi-Cluster Data"
> [2] Qian and Zhai (2013) "Robust Unsupervised Feature Selection"
> [3] Liu et al. (2014) "Global and local structure preservation for feature selection"
> [4] Li et al. (2012) "Unsupervised feature selection using non-negative spectral analysis"

---

### Author Response · Authors · 2018-10-30
**Clarifying typos**

We spotted three typos that could hinder the reader's comprehension :

-In the notations section , the unnormalized Laplacian should read "L = Delta - S" (instead of "L = M - S")

-At the beginning of page 3, in the formal background section, the first eigenvector should read : "Xi_0 = Delta^(1/2) 1" (instead of "Xi_0 = M^(1/2) 1 ")

-In appendix 1, the Laplacian score should read : "L_j = (1/sigma_j) * Sum_(i,k) (X[i,j] - X[k,j])S_(i,k)"

We apologize for these errors and hope those clarifications prove useful.

---

### Meta-Review · Area_Chair1 · 2018-12-17
**Unconvincing novelty and empirical results**

**Confidence:** 5
**Recommendation:** Reject

**Metareview:**

This  paper presents an LLE-based unsupervised feature selection approach. While one of the reviewers has acknowledged that the paper is well-written with clear mathematical explanations of the key ideas, it also lacks a sufficiently strong theoretical foundation as the authors have acknowledged in their responses; as well as novelty in its tight connection to LLE. When theoretical backbone is weak, the role of empirical results is paramount, but the paper is not convincing in that regard.